# An ultralight, tiny, flexible six-axis force/torque sensor enables dexterous fingertip manipulations

Qian Mao ⓞ, Zijian Liao ⓞ, Shiqiang Liu, Jinfeng Yuan ⓞ & Rong Zhu ⓞ ✉

Multi-dimensional mechanoreceptors are crucial for both humans and robots, providing omnidirectional force/torque senses to ensure dexterous and precise manipulations. Current six-axis force/torque sensors are bulky, heavy, and rigid with complicated sensing structures and high-cost manufacture. Although flexible force sensors have emerged recently, their perceptive dimension and performance are limited and still unsatisfactory for practical applications. Here, we propose an ultralight (0.30 g), tiny (fingertip size), and flexible six-axis force/torque sensor with a simple structure and low-cost fabrication. The sensor accurately perceives six-dimensional force/torque via capturing the spatial strain field of an elastic piezo-thermic material utilizing web-like scattered thin-film thermoreceptors. Integrating the sensor on the fingertip of humans or robots, we or robots can dexterously manipulate objects (e.g., open bottle cap), play games, and accomplish human-robot collaborative operations via easy fingertip-touch, demonstrating a broad prospect in applications of helping disabled and elderly people, intelligent robots, and virtual reality.

Intelligent robots have been widely demanded in many application fields such as industrial manufacturing[1,2], medical services[3], and home services[4–6]. The advancement of robot technology has also driven the growing demand for smart tactile sensing. Among them, flexible multi-dimensional force sensing on fingertips is particularly essential. When a robot executes fine operations such as unscrewing bottle caps, it needs to perceive not only normal force but also shear force. Multi-dimensional force sensors can enrich the robot's perception capabilities, help fine operation[7–10], stable grasping[4], etc. Nowadays, force sensors can be divided into piezoresistive[3,11–14], piezocapacitive[15–21], triboelectric[7,22–24], piezoelectric[8,25,26], magnetic[4,27], visual-tactile[6,28–31], and optical fiber[10,32–34] types. To realize multi-dimensional force sensing, the sensors usually employ special microstructures[5,14,18,23,35] (such as hemispheres, prism structures) or multiple arrays[8,17,19,22,36], most of which are complicated and hard to fabricate.

The perception of three-dimensional force is not enough in many application scenarios, it usually needs to further include the perception of three-dimensional torque[4,27,37,38]. In fact, human skin has the ability to perceive six-dimensional force/torque, offering our hands to perform delicate manipulations. Humans can perform fast, robust, and dynamic manipulation tasks with their fingers. The fingertips are recognized to be the most delicate parts, allowing them to press, roll, and/or slide during manipulation. The multi-dimensional force/torque sensing on fingertips capably provides humans or robots a strong maneuverability, even for those with limited mobility, such as disabled and elderly people.

Therefore, six-dimensional force/torque sensing on the fingertips is essential[11]. However, current six-axis force/torque sensors are bulky, heavy, and rigid with complicated sensing structures and high-cost manufacture[20,39–41]. The flexible force sensors have insufficient perceptive dimension (e.g., only one to three axes) and unsatisfactory

State Key Laboratory of Precision Measurement Technology and Instruments, Department of Precision Instrument, Tsinghua University, Beijing, China.
✉e-mail: zr_gloria@mail.tsinghua.edu.cn

performances in measuring range and accuracy[5,19,42], which limits their practical application for fingertip manipulations.

Here, we propose a flexible six-dimensional force/torque sensor with a simple structure, a small size (12 mm × 15 mm × 5 mm, fingertip size), and an ultralight weight (0.30 g) (Fig. 1a and S1). The sensor is comprised of two sensing layers sandwiching a silver-doped porous polydimethylsiloxane (PDMS) layer (Fig. 1b). The sensing layer is structured with the scattered thin-film thermistors fabricated on a flexible polyimide. The sensor perceives the force/torque via capturing the spatial strain field of an elastic piezo-thermic material (i.e., doped porous PDMS) using web-like scattered thin-film thermistors on the top and bottom (Fig. 1c). The piezo-thermic material transduces the strain to the thermal conductance change of its own material[43]. The thin-film thermistor serves as a thermoreceptor that is electrically heated and detects the heat-conducting change of the piezo-thermic material to capture the spatial strain (Fig. 1d). In virtue of highly sensitive strain detection based on this thermosensation, the sensor achieves a wide measuring range and a high accuracy in the measurements of the six-axis force/torque. The sensor is flexible, compact, ultralight, and particularly fits with fingertip use. We further showcase that humans can handily maneuver vehicles by his/her fingertip-touch, and thus potentially make up for the lack of mobility of the disabled and elderly people (Fig. 1e). Robots equipped with the sensor can dexterously manipulate objects like humans, e.g., uncapping a medicine bottle with a safe lock.

## Results

The six-axis force sensor is comprised of a top sensing layer, a bottom sensing layer, and a silver-doped porous polydimethylsiloxane (PDMS) in the middle (Fig. 1b). The top and bottom sensing layers have the same structure containing scattered thin-film thermistors composed of chromium (Cr, 30 nm thick) and platinum (Pt, 120 nm thick) fabricated on a flexible polyimide (detailed structure in Supplementary Fig. S2). As shown in Fig. 1c, we propose a six-axis force/torque sensor utilizing a web-like sensing architecture to capture the spatial strain field induced by external force/torque stimuli. The spatial strain field

of an elastic material can also map external force/torque stimuli. As shown in Fig. 1c, four scattered red thermistors with small resistances (about 50 Ω each) are electrically heated and denoted as hot-films, and four blue thermistors with larger resistances (about 300 Ω each) are unheated and denoted as cold-films. A hot-film and an adjacent cold-film constitute a sensing unit and are connected to a constant temperature difference (CTD) circuit (Fig. 1c and S3). The hot-film is heated to a set temperature (a constant 10 °C higher than ambient temperature) serving as both heater and temperature sensor, and transfers the heat with the surrounding porous PDMS (Fig. 1d). A force/torque induces a spatial strain of the porous material, altering its thermal conductance that changes the thermal field of the hot-film and then detected by the hot-film. This thermosensation-based strain detection has extremely high sensitivity. The cold film is unheated (served as an environment temperature sensor) and also utilized to make temperature compensation for the hot-film via the CTD circuit to eliminate the influence of ambient temperature. The detailed temperature compensation principle of the sensor is in Supplementary Notes. The power consumption of the sensor is about 80 mW.

Figure 2a shows the schematic diagram of the sensor under different force stimuli: null force, $F_z$ (normal force), $F_y$ (shear force), $M_x$, and $M_z$. When null force is applied, the middle porous PDMA is uncompressed. The scattered hot-films transfer the heat to the surrounding porous PDMS and reach the thermal equilibrium. When a normal force $F_z$ is applied, the middle porous PDMS is compressed and deformed, which alters the thermal conductivity of the porous PDMS because the proportion of air within is changed. As shown in Fig. 1d, the spatial strain in the porous PDMS generates a corresponding thermal conducting change in the porous material that is detected by the hot-film underneath the porous material. In this way, the spatial strain field of the middle porous material can be captured by the scattered hot-films on the top and bottom layers. When a shear force $F_y$ is applied, the induced strains of the porous PDMS on the left and right sides are asymmetric, which are captured by the scattered hot-films as well. Similarly, $F_x$, $M_x$, $M_y$, and $M_z$ generate their own strain fields and are detected by the scattered hot-films using the same sensing

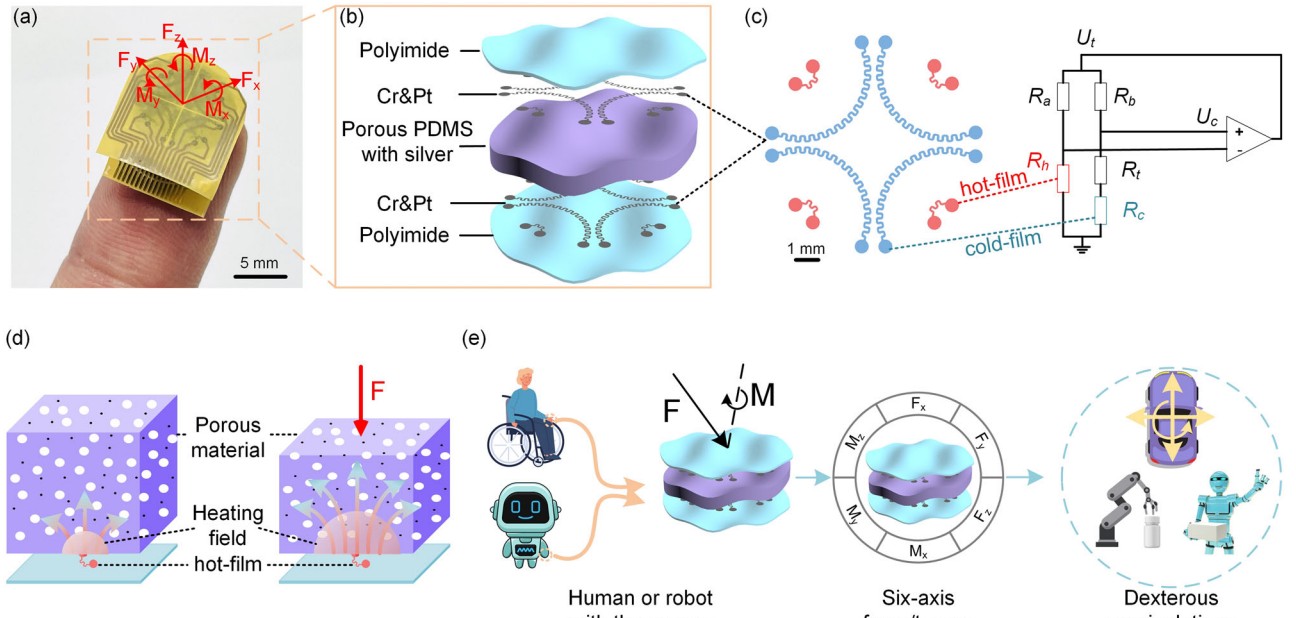

**Fig. 1 | An ultralight, compact, flexible six-axis force/torque sensor. a** The photograph of the six-axis force/torque sensor on a fingertip. **b** Detailed structure of the six-axis force/torque sensor. **c** Diagram of sensing units and conditioning circuit, where the scattered red parts are hot films and the blue parts are cold films. **d** Schematic diagram of the working principle of perceiving spatial strain based on thermosensation. The force-induced strain generates the thermal field change that is detected by the hot-film. **e** The six-axis force/torque sensor can be applied to dexterous manipulations for humans and robots.

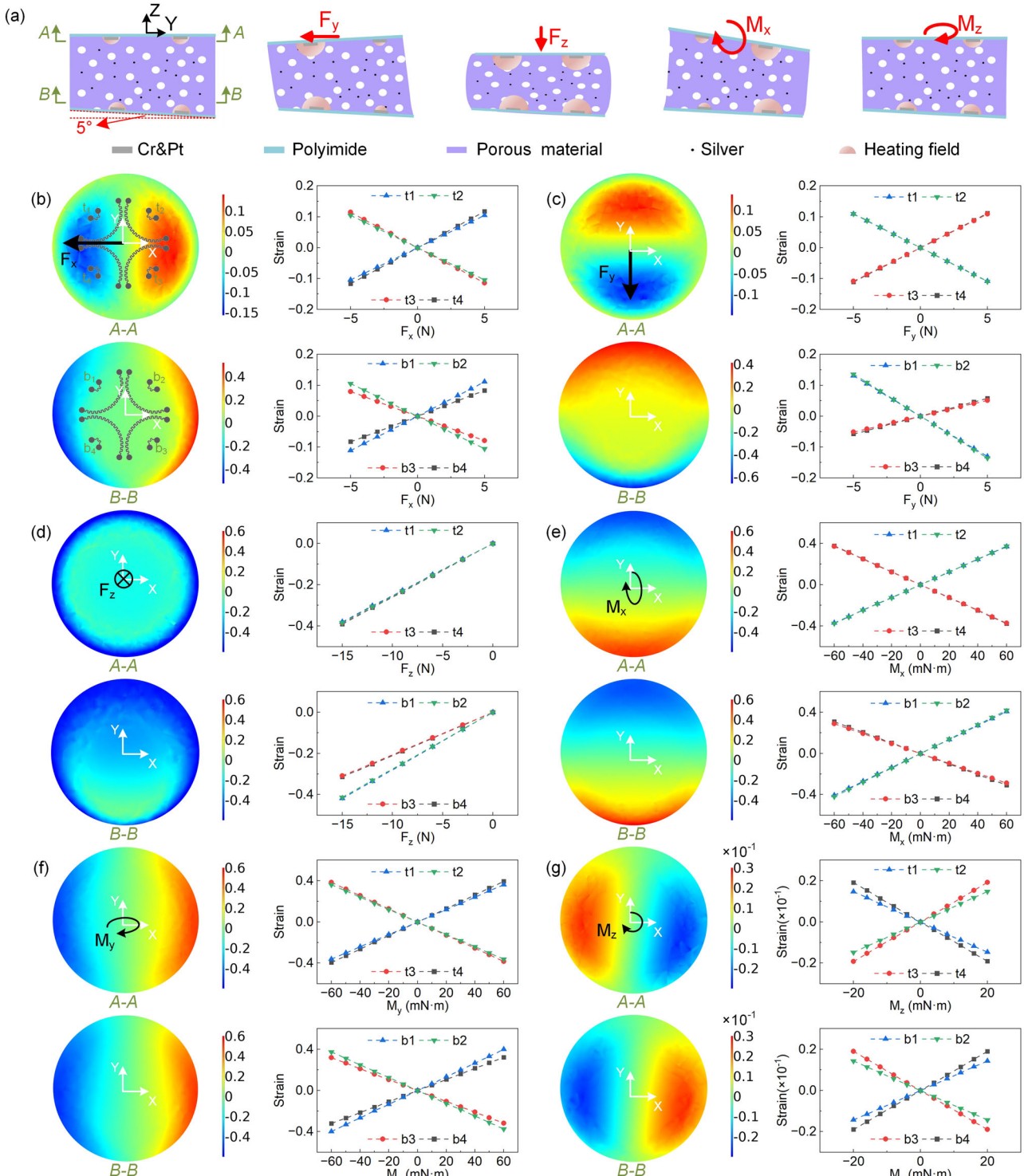

**Fig. 2 | The working principle and the simulation results of the sensor.**
**a** Schematic diagram of the deformation and heat transfer of the sensor with null force, $F_y$, $F_z$, $M_x$, and $M_z$ stimuli. **b**–**g** Left: the strain cloud diagram of the A-A plane and the B-B plane. Right: the simulation results of the relationship between the strains at the sensing units on the top and bottom layers and the applied force/torque.

principle. As a consequence, the output voltages of the scattered hot-films respond to the six-axis force/torque stimuli.

It is worth mentioning that we tailor a slight bevel (5° in this work, detailed parametric optimization in Supplementary Fig. S4) between the top and bottom sensing layers to construct the distinguishable spatial strain fields in the porous PDMS for identifying six-dimensional forces and torques (details in Fig. 2a). In order to explain the sensing

principle in more detail, we simulate the strain fields of the sensor under different force and torque stimuli. Figure 2b–g show the strain cloud diagram in two cross-sectional planes (A-A and B-B, marked in Fig. 2a) and the strain changes at the sensing units on the top layer (denoted as $t_1$, $t_2$, $t_3$, $t_4$ in Fig. 2b) and the bottom layer (denoted as $b_1$, $b_2$, $b_3$, $b_4$ in Fig. 2b) under force/torque stimuli. As shown in Fig. 2b, when the shear force $F_x$ is applied, both strains in the A-A plane and the

B-B plane show an antisymmetric pattern along the Y axis. When the shear force $F_y$ is applied, the strains show an antisymmetric pattern along the X axis (Fig. 2c). When the normal force $F_z$ is applied, the strains at the top sensing layers and the bottom sensing layers show a little difference because of the bevel (Fig. 2d). When the torque $M_x$ is applied, the strains on the top and bottom sensing layers are almost the same, showing an opposite pattern along the X axis (Fig. 2e). When the torque $M_y$ is applied, the strains show a similar pattern with $M_x$ but along the Y axis (Fig. 2f). When the torque $M_z$ is applied, due to the bevel between the top sensing layer and the bottom sensing layer, the strains in the A-A plane and the B-B plane are opposite, and also show an antisymmetric pattern along the Y axis (Fig. 2g). Therefore, the six-axis forces and torques induce the discernable strain fields in the porous material, which are captured and recognized by the scattered hot-films at the top and bottom sensing layers.

We fabricate the scattered thin-film thermistors (Pt/Cr) on a flexible printed circuit board (FPCB, polyimide) using the photo-lithography and sputtering. The porous silver-doped PDMS is fabricated using a sacrificial template method and adhered between the top and bottom sensing layers. The detailed fabrication process can be found in Materials and Methods, and Supplementary Fig. S5. The manufacturing process is very simple and low-cost. The fabricated sensor has a fingertip size and an ultralight weight of 0.3 g (shown in Fig. 1a and S1).

We conduct the experiments to verify the six-axis force/torque sensor. The testing system is shown in Supplementary Fig. S6. Supplementary Fig. S7 shows the response results of the bottom sensing layer (denoted as $Ub_1$, $Ub_2$, $Ub_3$, $Ub_4$) and the top sensing layer (denoted as $Ut_1$, $Ut_2$, $Ut_3$, $Ut_4$) under different force/torque stimuli.

To figure out the six-axis forces and torques from the output voltages of the top and bottom hot-films, we conduct a data fusion utilizing a convolutional neural network (CNN). The voltage responses of the top and bottom hot-films in the sensor are served as the inputs of the CNN, and the six-axis force/torque quantities are served as the outputs, as shown in Fig. 3a. The CNN contains two hidden layers, with 20 and 10 neurons in the first and second layers, respectively.

Figure 3b−g show the detailed measurements of our six-axis force/torque sensor. The measuring ranges of $F_x$, $F_y$, $F_z$, $M_x$, $M_y$, and $M_z$ cover $-5 \sim 5$ N, $-5 \sim 5$ N, $-15 \sim 0$ N, $-60 \sim 60$ mN·m, $-60 \sim 60$ mN·m, and $-20 \sim 20$ mN·m, respectively. It is seen that the measurement results of $F_x$, $F_y$, $F_z$, $M_x$, $M_y$, and $M_z$ are in good agreement with the actual values. It is estimated that the correlation coefficients ($R^2$) between the actual values and the measured values reach 0.993, 0.995, 0.999, 0.999, 0.999, and 0.998 for $F_x$, $F_y$, $F_z$, $M_x$, $M_y$, and $M_z$, respectively. The experimental results also show that the root mean square error (RMSE) of the measured $F_x$, $F_y$, $F_z$, $M_x$, $M_y$, and $M_z$ are 0.18 N, 0.15 N, 0.18 N, 0.58 mN·m, 0.62 mN·m, and 0.26 mN·m, respectively. Furthermore, the crosstalk errors among different forces and torques are tested and show relatively low, which ensures simultaneous and independent perceptions of six-axis force/torque (Supplementary Fig. S8). The detection limits of $F_x$, $F_y$, $F_z$, $M_x$, $M_y$, $M_z$ are 0.052 N, 0.05 N, −0.004 N, 1.20 mN·m, 1.22 mN·m and 0.31 mN·m, respectively (Supplementary Fig. S9). The response time and recovery time are both 90 ms (Fig. 3h). Furthermore, we test the repeatability of the sensor in 100,000 cycles of −15 N (normal force) (Fig. 3i), the results show the sensor has good durability. To validate the environment temperature and humidity immunity of the sensor, we test the sensors' responses at different temperatures and humidites, the results indicate that the sensor is almost unaffected by environment temperature and humidity (Fig. 3j, details in Supplementary Figs. S10 and S11). In addition, the topologically similar shapes of the hot-film and cold-film ensure the sensor free from bending influence[44] (principle in Supplementary Notes, Supplementary Fig. S12). All the above results show that the sensor has a low detection limit, a fast response, good repeatability, and excellent stability. The test results evidence that the

sensor enables accurate measurements of six-axis force/torque in a wide measuring range, which ensures delicate manipulation by fingertips. Figure 3k gives the comparison of our six-axis force/torque sensor to existing flexible multi-axis force sensors reported by others (more comparisons in Supplementary Table S1). The axes in the radar diagram indicate the force range (normal force and shear force) or the inverse of the RMSE of the sensors, where the larger coverage indicates the better performance of the sensor. It is seen that our six-dimensional force/torque sensor is significantly superior to other multi-dimensional force sensors, exhibiting more dimensions and functionalities, higher accuracy, and wider measuring range.

Flexible fingertip sensors with perceptions of six-dimensional force/torque can provide robots with great capability of dexterous operation, like humans. We equip the six-axis force/torque sensor on the robotic fingertip to deal with the complex manipulation tasks, e.g., uncapping medicine bottles. In our daily lives, medicine bottles are usually with safety locks. In order to prevent the medicine from children, the cap needs to be pushed down and then turned to open, as shown in Fig. 4a. For the robotic uncapping bottle, in process (i), the robotic hand does not yet touch the bottle cap (shown in Fig. 4b), and the response signals of the sensor are null (as shown in Fig. 4c). In process (ii), the robotic hand starts to grasp the bottle cap and hold steady. Correspondingly, the normal force $F_z$ increases and tends to stable. In process (iii), the robotic hand pushes the bottle cap downward (defined as X axis), correspondingly, the shear force $F_x$ increases. In process (iv), the robotic hand rotates the bottle cap, and the shear force $F_y$ increases accordingly. Afterward, as the bottle cap is unscrewed, the robotic hand releases the bottle cap, and the three-axis force signals of the sensor return to null. The detailed robotic uncapping process is shown in Supplementary Movie S1. Obviously, the sensor on the robotic fingertip can accurately measure the multi-axis forces during the robot's uncapping bottles, which would greatly facilitate robots for dexterous operations.

Flexible fingertip sensors with perceptions of six-axis force/torque can also assist humans (especially disabled people) in controlling various vehicles via easy fingertip-touch. For severely disabled people, steering electric wheelchairs is generally difficult. Using the six-axis force/torque sensor, disabled people can easily and flexibly steer the wheelchair with just one finger, which provides great convenience. We simulate a scenario in which a subject plays a tank game via a force/torque sensor on his fingertip (Supplementary Movie S2). In this case, the player touches the force/torque sensor with his fingertip to control the tank. The tank has multiple control modes, including: moving forward, moving backward, body turning left, body turning right, gun barrel turning left, gun barrel turning right, and firing. We use $F_y$ to control the forward and backward movement of the tank. In process (i), when $F_y > 0$, the tank moves forward; in process (ii), when $F_y < 0$, the tank moves backward. In addition, we use $F_x$ to control the tank's body to turn left or right. In process (iii), when $F_x < 0$, the tank body turns left, and in process (iv), when $F_x > 0$, the tank body turns right; in processes (v) and (vi), we use $M_y$ to control the barrel to turn left or right. When $M_y < 0$, the barrel turns left, and when $M_y > 0$, the barrel turns right; finally, in process (vii), when the normal force $F_z > 10$ N, we trigger the tank to fire. Since the six-axis force/torque sensor has the capability to realize multimodal control of the tank, there is a good intuitive mapping between the real tactile perception and the virtual tank movement. The virtual exercises through the sensor are expected to be applied to the actual training scenario program.

Furthermore, we apply the six-axis force/torque sensor to help a disabled people who lose their limb motion capability. People can control a robot via easily moving his/her fingertip on the six-axis force/torque sensor to tidy up life items (Supplementary Movie S3). Figure 5a shows a schematic diagram of this control. We correspond the output of the six-dimensional force to the six-degree-of-freedom motion of the robotic arm. For example, when $F_x > 0$, the robot moves forward in

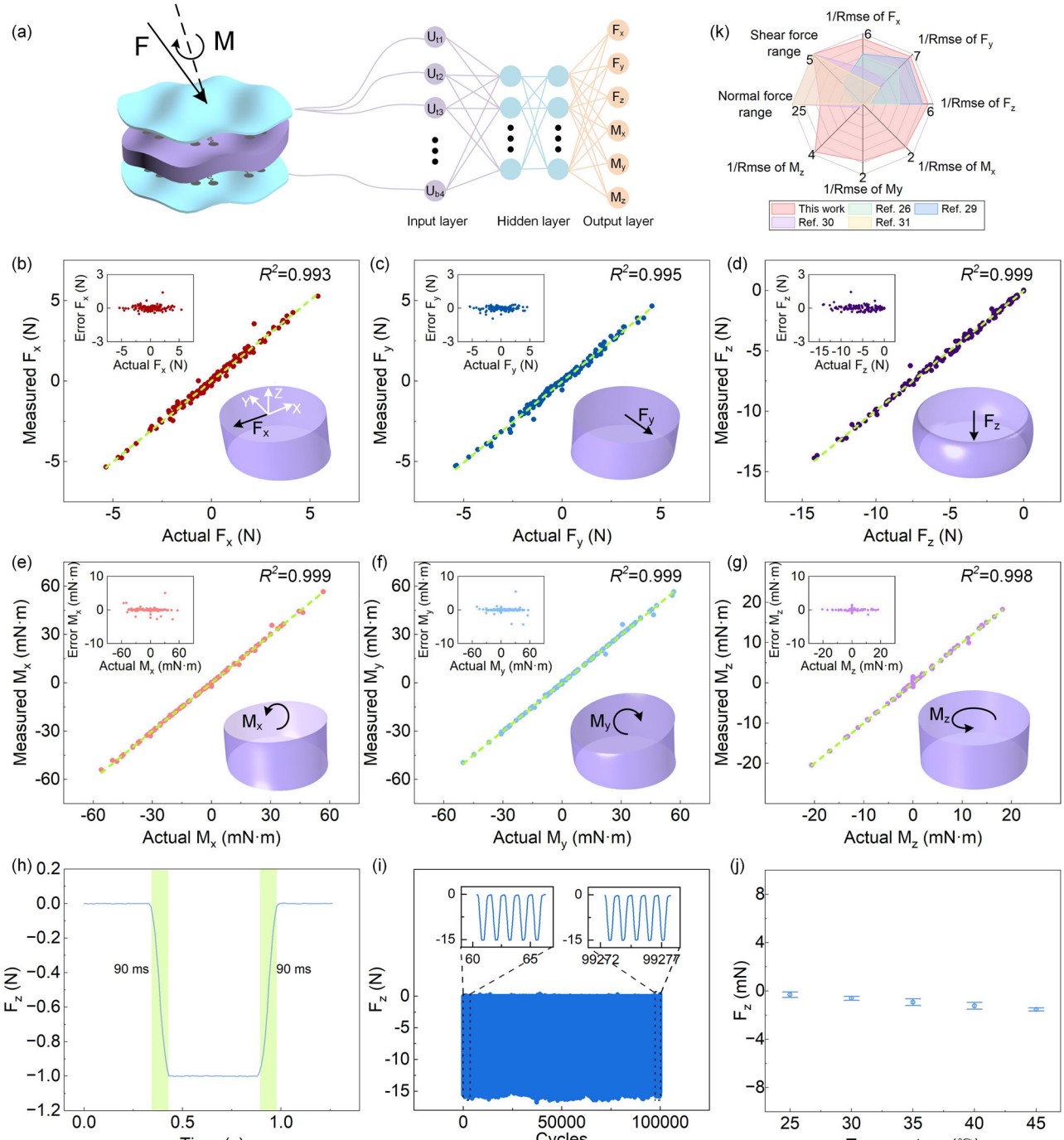

**Fig. 3 | Performances of the six-axis force/torque sensor. a** Data fusion diagram for figuring out the six-axis forces and torques. **b** Performance of the sensor for $F_x$. **c** Performance of the sensor for $F_y$. **d** Performance of the sensor for $F_z$. **e** Performance of the sensor for $M_x$. **f** Performance of the sensor for $M_y$. **g** Performance of the sensor for $M_z$. **h** Response time of the sensor. **i** The repeatability of the sensor under 100,000 cycles with −15 N (normal force) loading and unloading. **j** The temperature influence on the force sensor (the details in Supplementary Fig. S10), error bars are standard deviations of five repeated measurements of one specific sensor. **k** Radar diagram of the performance of our six-axis force/torque sensor compared to other flexible multi-axis force sensors.

the X - direction. Conversely, when $F_x < 0$, the robot moves backward in the *X*-direction. The motion control on other axes is similar. For the control of the robotic hand, when the sensor detects two consecutive taps, the robotic hand grasps or releases the object. In this way, people enable coordinated control of the robotic arm and robotic hand via an easy fingertip touch, and further accomplish robotic housekeeping.

Figure 5b shows the response signals of the force/torque sensor in controlling a robot to tidy up some items on a desktop. In process (i), the sensor detects $F_y < 0$ and $F_z < 6$ N, and the robot moves forward in the Y direction and backward in the Z direction; then, in process (ii), the sensor detects $F_x > 0$ and $F_z > 6$ N, the robot moves forward in the X direction and moves backward in the Z direction, and then the robot approaches the object (a cup). In process (iii), the sensor detects three-axis torques, and the robotic arm rotates along three axes to adjust its grasping posture; in process (iv), the sensor detects two consecutive taps, and the robotic hand grasps the object; in process (v), the sensor detects $F_z < 6$ N, and the robot arm moves forward in the Z direction; in process (vi), the sensor detects the

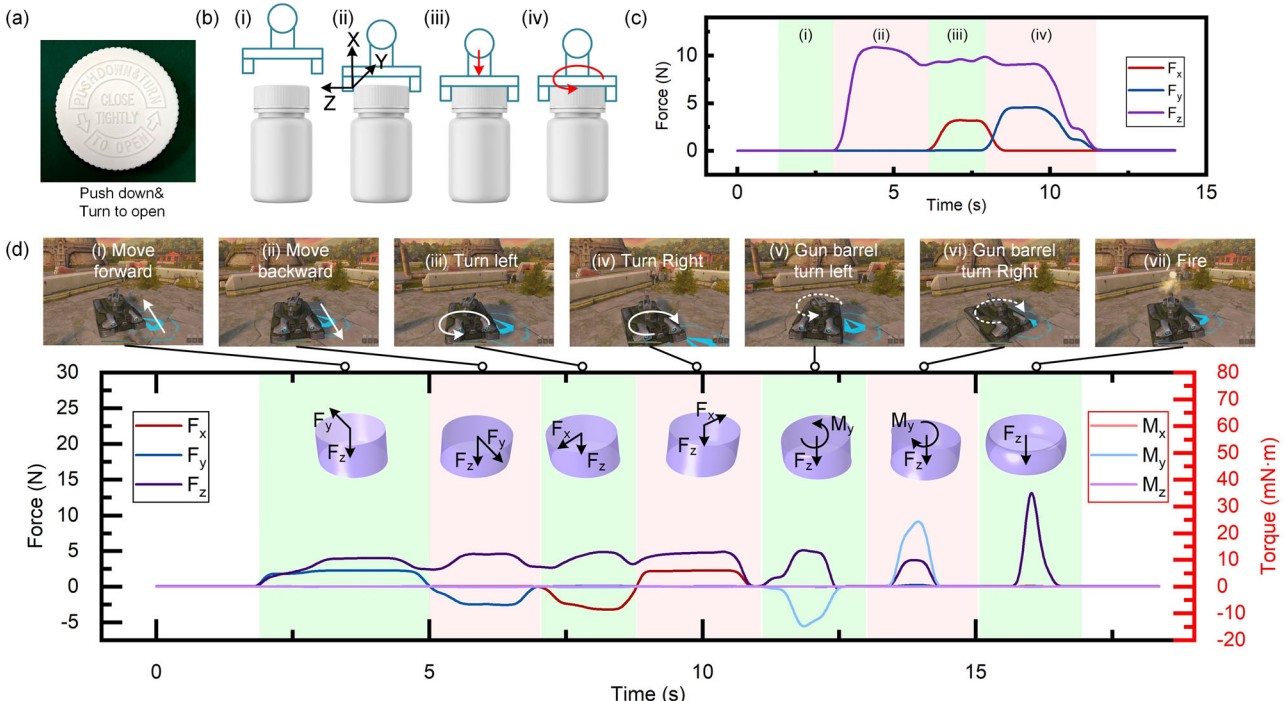

**Fig. 4 | Applications of the six-axis force/torque sensor. a** A bottle cap with a safety lock. **b** The schematic diagram of a robotic uncapping a bottle. **c** Real-time responses of the sensor during the robot opening the bottle cap. **d** Real-time responses of the six-axis force/torque sensor in controlling a tank game. The top images show the corresponding actions of the tank.

three-axis torques, and the robot arm rotates along three axes to adjust its posture; in process (vii), the sensor detects $F_y > 0$, and $F_z < 6$ N, and the robot arm moves horizontally to the target position (on the shelf); finally, in process (viii), the sensor detects two consecutive taps, and the robotic hand releases the cup and places it on the shelf. The detailed demonstration is shown in Supplementary Movie S3. Utilizing the smart force/torque sensor on a human's fingertip, disabled people can manipulate a robot flexibly and potentially regain self-care ability.

## Discussion

In this paper, we propose a flexible six-axis force/torque sensor with a simple and compact structure, fingertip size, ultralight (0.30 g), and low-cost fabrication. The sensor perceives six-axis force/torque via capturing the spatial strain of an elastic piezo-thermic material using the web-like scattered thin-film thermistors based on thermosensation. The sensor exhibits good performances with wide measuring ranges of $-5 \sim 5$ N, $-5 \sim 5$ N, $0 \sim 15$ N, $-60 \sim 60$ mN·m, $-60 \sim 60$ mN·m, and $-20 \sim 20$ mN·m for $F_x$, $F_y$, $F_z$, $M_x$, $M_y$, and $M_z$, and high accuracy of 0.18 N, 0.15 N, 0.18 N, 0.58 mN·m, 0.62 mN·m, and 0.26 mN·m (RMSE), respectively. The sensor also reaches a low detection limit (0.004 N), fast response (90 ms), and good repeatability (100,000 cycles).

The flexible force/torque sensor has a broad prospect of application in assisting humans and robots. We demonstrate the scenarios of robotic fine manipulation and human-robot collaborative operation. We equip the sensor on the robotic fingertip to assist in uncapping a medicine bottle with a safety lock. With the aid of multidimensional force sensing, a complicated process of applying force during uncapping bottles is accurately captured. We also apply the sensor to help humans. As the demonstration shows, people can flexibly steer and fire a tank game with easy fingertip-touch, showing its promising potential for virtual reality or remote control. We also showcase that the sensor assists disabled people in controlling a robot for housekeeping. With this tiny sensor on the fingertip, humans can flexibly operate robotic arms and robotic hands to tidy up daily items,

which may pave the way for disabled people to regain their self-care ability.

The functionality of the proposed six-axis force/torque sensor can be further expanded via integrating a thermosensation-based interface sensing layer on the top to endow the sensor with more perceptive capabilities of matter recognition (shown in Supplementary Fig. S13), slip detection, texture identification, etc. Such a multimodal tactile sensor is promising to achieve better human-robot interaction. For example, in a robotic dexterous manipulation, the robot needs not only six force/torque sensing, but also slipping detection and object recognition, so as to stably grasp objects and make fine sorting. The rich and delicate tactile sensing capability would definitely promote a great development of human-robot collaboration, embodied intelligence, etc.

## Methods

### Fabrication of the six-axis force sensor

The sensor consists of a top sensing layer, a bottom sensing layer, a silver-doped porous polydimethylsiloxane (PDMS) in the middle. The detailed fabrication process is as follows:

The top sensing layer and the bottom sensing layer have the same structure and the same fabrication process. As Supplementary Fig. S5a shows: (i)The sensing layer is fabricated on a flexible polyimide substrate. (AP8525R, DuPont Co. Ltd., Wilmington, America) (ii) The pads and wires are fabricated on the polyimide substrate by flexible printed circuit board (FPCB) technology. (iii) Spin coat 30 μm thick photoresist (KXN5735-LO, Rdmicro Co. Ltd., Suzhou, China) on the polyimide substrate and use lithography to obtain the pattern. (iv) The chromium/platinum (30 nm/120 nm) films are sequentially deposited by sputtering. (v) Soak it in acetone for 2 h to remove the photoresist and form the corresponding pattern. (vi) Deposit a parylene film with a thickness of 6 μm as a protective layer.

As Supplementary Fig. S5b shows, the silver-doped porous polydimethylsiloxane (PDMS) is fabricated by mixing silver nanoparticles (diameter is 60–150 nm, XFJ14, XFNANO Materials Tech Co.

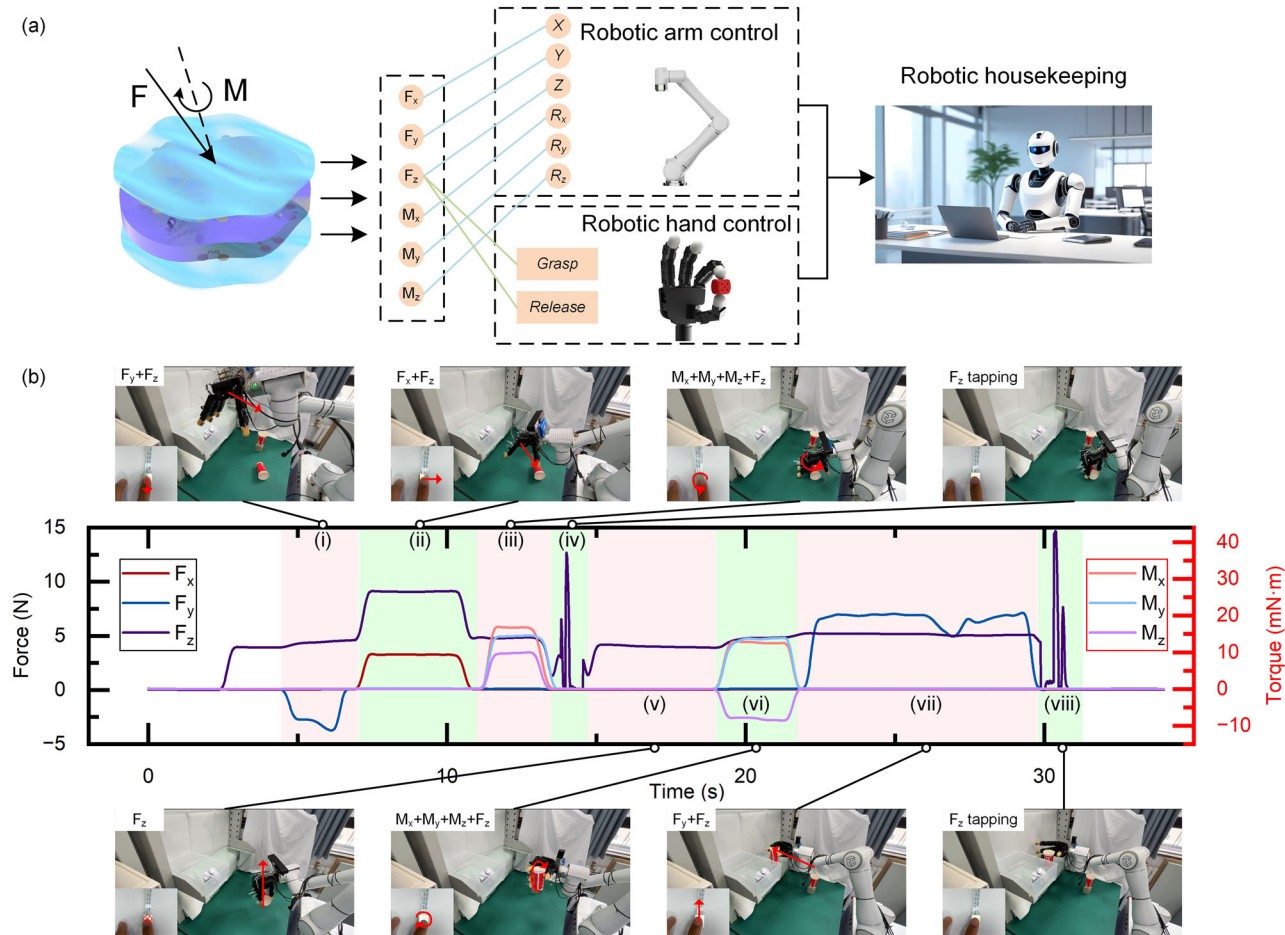

**Fig. 5 | Using the six-axis force/torque sensor to control a robot for housekeeping. a** The schematic diagram of controlling a robot using a six-axis force/torque sensor on the fingertip. **b** Real-time responses of the six-axis force/torque sensor in the robotic control process for tidying up daily items. The top and bottom images show the corresponding robot actions, and the bottom left insets show the corresponding fingertip-touch.

Ltd, Nanjing, China), prepared PDMS solution (Sylgard 184, Dow Corning Company, Wiesbaden, Germany, the ratio of base agent: cross-linker was 10:1 wt%), and citric acid monohydrate particles (CAM, Sinopharm Chemical Reagent Co. Ltd., Shanghai, China). The volume ratio of silver nanoparticles is 2.5 vol% (parametric optimization in Supplementary Fig. S14). The mass ratio of PDMS: CAM is 1:3.5. A blender (HMV200D, Hasai Technology Co. Ltd., Shenzhen, China) to blend the mixture thoroughly in three steps: (a) Revolution 900 rpm and rotation 600 rpm, 30 s. (b) Revolution 1500 rpm and rotation 600 rpm, 540 s. (c) Revolution 900 rpm and rotation 600 rpm, 30 s. Then the mixture is cured at 75 °C for 2 h in an acrylic mold. Then the mixture is immersed in ethanol for 24 h to dissolve the CAM to form the porous material. Finally, it is washed with deionized water and dried at 70 °C for 1 h. During this fabrication, if heating is not required, the fabrication is at room temperature (23 ~ 25 °C) and humidity (10% ~ 30%).

Furthermore, in order to ensure the touch stability and reliability in practical uses, we add a thin aluminum sheet onto the top sensing layer of the sensor as encapsulation.

### Calibration of the six-axis force sensor
The six-axis force calibration is conducted by applying force on the sensor using a mechanized z-axis stage (Handpi Co. Ltd., Yueqing, China) with a force gauge (SH-50, Sundoo Co. Ltd., Wenzhou, China, 0.01 N of resolution). The detailed device can be found in Supplementary Fig. S6. The sensor is placed on an adjustable angle inclined plane and can be rotated by a turntable. In this way, different three-axis

forces and three-axis torques can be applied. The dataset contains 418 examples, where 40% of the dataset is the training set, 10% of the data is the validation set, and 50% of the dataset is the test set. The neural network is trained using the Bayesian regularized back-propagation method, with the learning rate set to 0.01 and the batch size set to 167.

## Data availability
The data that support the findings of this study are available within the paper and the Supplementary Information. The source data generated in this study are provided in the Source Data file. Source data are provided in this paper.

## Code availability
The MATLAB code and data are available on GitHub at https://github.com/mq-0109/An-ultralight-tiny-flexible-six-axis-force-torque-sensor-enables-dexterous-fingertip-manipulations and Zenodo https://doi.org/10.5281/zenodo.15553350[45].

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

## Acknowledgements

Beijing Natural Science Foundation (Grant No. L247001, R.Z.). National Natural Science Foundation of China (Grant No.51735007, R.Z.). National Natural Science Youth Foundation (Grant No. 62401333, J.F.Y.). China Postdoctoral Science Foundation (Grant No. 2023M742014, J.F.Y.).

## Author contributions

Conceptualization: R.Z. Methodology: Q.M. and R.Z. Investigation: Q.M. and Z.J.L. Visualization: Q.M., Z.J.L. and J.F.Y. Funding acquisition: R.Z. and J.F.Y. Project administration: R.Z. Supervision: R.Z. and S.Q.L. Writing – original draft: Q.M. Writing – review & editing: Q.M. and R.Z.

## Competing interests

Authors declare that they have no competing interests.
