## [Transparent Peer Review file · Nature Communications]

An ultralight, tiny, flexible six-axis force/torque sensor enables dexterous fingertip manipulations

Corresponding Author: Professor Rong Zhu

Version 0:

Reviewer comments:

Reviewer #1

(Remarks to the Author)

In this manuscript, the authors reported a flexible fingertip sensor using convolutional neural network to decouple six-dimensional force/torque, which further can be used for game playing and robotic manipulation. However, some descriptions of the sensor are not entirely appropriate, and there are still certain details that require further elaboration. Thus, I would like recommend consider major revision before publication.

1. The author claims that the sensor is “ultralight, tiny with a simple structure”, but it seems inappropriate or meaningless. First, attention should be paid to the mass of the overall sensing system. The 0.3 g weight of the front-end detection device not only lacks comparison with other devices, but also holds little significance for interactive applications. Second, a size of 12 mm×15 mm×5 mm for a fingertip sensor is not particularly small when compared to other reported works. If the miniaturization of the sensor size is limited by the thermal field interferences between adjacent sensing units? Last, the composite sensor is actually an integration of 8 sensing units with 32 leads, which may make the detection circuit complicated. And if it expands to be a sensing array, the structure will be more complicated.
2. The author mentions that the structure of the sensor is a biomimicry of the scattering structure of a spider web, but this description seems somewhat far-fetched. Apart from the similarity in shape, there appears to be little connection between the sensor's functionality and that of a spider web.
3. The author designed a 5° tilt angle on the bottom of the sensor to “construct distinguishable spatial strain fields within the porous PDMS, aimed at identifying six-dimensional forces and torques”. However, it appears that even without the tilt, the strain fields in various regions could still be well differentiated. Please give a more detailed elaborate on the function of the tilt. Additionally, when M_z is applied (Fig. 2g), why does the principal strain exhibit a symmetrical distribution rather than forming a circular pattern aligned with the load direction? Please discuss
4. The author should further refine the calibration results of the sensor. In the supplementary tests shown in Fig. 3, why was only F_z tested? What about the responses of other forces or torques? Additionally, the response time test in Fig. 3h is not standardized when the change of F_z is less than 1% FS, which can not accurately reflect the dynamic characteristics of the sensor in actual use. Finally, in Fig. 3k, the coordinates lack numerical values, and most of the comparison subjects are not six-dimensional force sensors.
5. During the controlling experiments of the tank and the robotic arm, the operator merely uses their fingers to manipulate the sensor. Typically, multiple forces or torques would be applied simultaneously. However, the signals detected often respond to only specific types of forces or torques. Is this due to the operator's exceptionally precise hand control? When training the neural network, did the author use manual input or a calibration platform to generate the input signals? If the latter was used, then based on the platform setup shown in Fig. S5, it appears that the platform may not accurately measure the magnitude of torques.
6. At the end of the article (Fig. S9), the author suggests that a laminated design could be employed to detect different objects. However, the newly added components are entirely a reuse of the previous work, which lacks innovation and further

complicates the sensor structure.

Reviewer #2

(Remarks to the Author)

The manuscript proposes a flexible six-axis force/torque sensor via capturing the spatial strain field of an elastic material utilizing scattered thin-film thermoreceptors. The sensor is ultralight (0.3 g), tiny, simple-structured, and the sensing performance is great in perceiving six-dimensional force/torque. The sensor particularly fits with the fingertip use, allowing dexterous fingertip manipulations. The manuscript also showcases some demonstrations of human-machine interaction tasks using the sensor. The work is interesting, the sensing design is novel, and the application is promising. There are minor issues that need to be addressed:

1. How does the environment temperature influence the sensor? How to realize temperature compensation?
2. How about the humidity influence to the sensor?
3. According to the coordinate system defined in Figure 3b, I think F_z in Figure 3h~3j should be <0 . How many repeat measurements were taken in the experiment of Figure 3j? How were the error bars calculated?
4. How to train the data fusion model for figuring out six forces and torques needs to be described in more detail.
5. What is the power consumption of the sensor?
6. The detailed operation of robot in the application of helping disabled people should be described more clearly.
7. In the first paragraph of Introduction, the piezoelectric principle also needs to be included for force sensors.

Reviewer #3

(Remarks to the Author)

This work presents a flexible tactile sensor achieving 6-axis force/torque measurement through a thermoelectric-coupled design, addressing the challenge of high-dimensional signal decoupling measurement, demonstrating potential in delicate robot manipulation and human-mechanism interfaces. The study insufficiently validates flexibility, but is limited to flat-surface testing, lacks spatial resolution for multi-point force detection contradicting spiderweb-inspired claims, and omits many details of parametric optimization. Durability and environmental robustness remain unproven. Mandatory revisions are required to meet the publishing requirement including (1) sensor characterization under bending conditions (2) validation of localized/multi-force discrimination akin to spiderweb mechanics, (3) systematic parameter studies, and (4) expanded durability/environmental testing. Therefore, a major revision is recommended – methodological gaps must be resolved to solidify scientific contribution and translational value.

Q1. What are the noteworthy results?

1. This paper presents a flexible 6-axis force sensor with compact dimensions that simultaneously measures triaxial forces (F_x , F_y , F_z) and torques (τ_x , τ_y , τ_z). This resolves the contradiction between high flexibility and high-dimensional signal decoupling in measurement.
2. The transformed mechanical stimuli into anisotropic thermal conduction signals (via Ag-doped porous PDMS), enabling unique responses to complex loads without bulky embedded electronics.
3. Demonstrated the implementation of the sensor into the robotic gripper's force sensing and remote robot control. The control of the gaming tanker and robotic arm are similar applications. The only difference is the first is a virtual robot while the latter is a physical robot.

Q2. Will the work be of significance to the field and related fields?

1. This work presents a flexible 6-axis force sensor. The proposed sensor also has the potential to provide a new way for human-machine interaction.

Q3. How does it compare to the established literature? If the work is not original, please provide relevant references.

1. There are two concerns: the "flexible" claim is insufficiently supported by the experiments and analysis, and the spiderweb-inspired spatial perception is not mechanistically connected to the thermal-resistive design.

Q4. Does the work support the conclusions and claims, or is additional evidence needed?

1. Insufficient experiments: Force/torque tests were conducted on flat surfaces (Fig. 3a-b). However, no data on performance under bending conformal states are presented (e.g., wrapping around 10 mm radius curvatures), which is essential to validate "flexible" claims. The sensor's thermoelectric response may degrade significantly under non-planar deformation due to altered pore geometries and Ag particle connectivity. Therefore, more experiments and analysis under bending conditions are required.
2. Overstated bio-inspiration: While spiderwebs excel in localized force detection and multi-point vibration discrimination, the current sensor only provides global 6-axis force estimation. There is no evidence of spatial resolution or multi-region independent sensing. It is suggested to apply two separate forces (e.g., F_x at the left edge and F_y at the right edge) to check if the sensor can resolve their locations and magnitudes simultaneously.
3. Not detailed parametric optimization: No theoretical/experimental proof for selecting a 5° title angle over other angles (e.g., 10°), and the justification for silver-doped porous polydimethylsiloxane, such as 2.5 vol% doping, is Vague.
4. Incomplete model validation: CNN Generalization: Training/validation data (60/40 split) lack an independent test set. Crosstalk errors under combined loads (e.g., $F_x + \tau_z$) are unexplored.
5. Insufficient durability test: 1000 cycles of 2 N loading (Fig. S3) are insufficient for medical/industrial use (>100,000 cycles needed per ISO 13485). Why not 5N and 15N, as the force-sensing range is 5N and 15N?

Q5. Are there any flaws in the data analysis, interpretation, and conclusions? Do these prohibit publication or require

revision?

Add more comparisons with other 6DOF sensors e.g., elastomeric strain gauges or FBG references.

Q6. Is the methodology sound? Does the work meet the expected standards in your field?

Extension works are required as stated in Q4.

Q7. Is there enough detail provided in the methods for the work to be reproduced?

1. Insufficient Protocol Details: Porous PDMS Fabrication: "Saccharine templating" (Methods) lacks critical parameters (e.g., stirring rate, humidity control). It is suggested to provide a step-by-step video protocol for PDMS-Ag layer synthesis.

2. Neural Network Training: Missing hyperparameters (batch size, optimizer settings), raising barriers to replication. Open source the CNN codebase and raw calibration data will be appreciated.

Reviewer #4

(Remarks to the Author)

Version 1:

Reviewer comments:

Reviewer #1

(Remarks to the Author)

The author has addressed all my comments, thus this paper can be accepted.

(Remarks on code availability)

Reviewer #2

(Remarks to the Author)

The authors have revised the manuscript carefully according to the reviewer's comments. The quality of this work is greatly improved. The reviewer is satisfied with the revision and has no more other suggestion for this work. I think that this manuscript should be accepted by the journal of Nature Communications.

(Remarks on code availability)

Reviewer #4

(Remarks to the Author)

(Remarks on code availability)

Please carefully check Fig. 3 (j). The results in this version and the previous version are opposite. Why? Please detail the reason.

Dear Reviewers,

Thank you very much for your comments and efforts contributed to our paper. We have carefully investigated all comments. They are very helpful to improve the manuscript. We believe this has led to the substantial revision of the paper. The responses to these comments are listed in the following and all of the revisions have been marked in **red font** in the text of the revised manuscript and Supplementary Information.

COMMENTS TO AUTHOR:

Reviewer #1

In this manuscript, the authors reported a flexible fingertip sensor using convolutional neural network to decouple six-dimensional force/torque, which further can be used for game playing and robotic manipulation. However, some descriptions of the sensor are not entirely appropriate, and there are still certain details that require further elaboration. Thus, I would like recommend consider major revision before publication.

Comment #1

The author claims that the sensor is “ultralight, tiny with a simple structure” , but it seems inappropriate or meaningless. First, attention should be paid to the mass of the overall sensing system. The 0.3 g weight of the front-end detection device not only lacks comparison with other devices, but also holds little significance for interactive applications. Second, a size of 12 mm×15 mm×5 mm for a fingertip sensor is not particularly small when compared to other reported works. If the miniaturization of the sensor size is limited by the thermal field interferences between adjacent sensing units? Last, the composite sensor is actually an integration of 8 sensing units with 32 leads, which may make the detection circuit complicated. And if it expands to be a sensing array, the structure will be more complicated.

Response #1

Thank you very much for your comments.

First, we have added a comparison with other devices in Table S1 in the Supplementary Information, and we can see that our sensor weighs much less than the others. For interactive scenarios, we think a lighter sensor can reduce the burden on the finger, thereby enabling more flexible interaction control for humans and robots. The front-end sensor is connected to the back-end circuit through the leads, so the weight of the back-end circuit does not affect its portability. Second, the sensor size of 12 mm×15 mm×5 mm is tailor-designed to fit the dimension of a finger, which facilitates dexterous force/torque manipulations on the finger. The sensor can be miniaturized further, and the thermal interferences can be reduced by downsizing the sensing unit and lowering the heating temperature. Last, if it expands to be a larger sensing array, we can use an array scanning circuit to minimize the lead complexity.

The corresponding revision can be seen on Page 18 in the revised Supplementary Information.

Comment #2

The author mentions that the structure of the sensor is a biomimicry of the scattering structure of a spider web, but this description seems somewhat far-fetched. Apart from the similarity in shape, there appears to be little connection between the sensor's functionality and that of a spider web.

Response #2

Thanks for your comment.

Spiders determine the location of prey on their webs by sensing spatial vibrations in their webs. Our sensor detects spatial strains by web-like sensing elements to determine the six-axis force/torque. The similarity is that they both perceive the spatial field of vibrations/strains to determine the target parameter. To avoid the confusion of biometric comparison, we have eliminated the biomimicry expression (Fig. 1c) and modified the description as “The sensor perceives the force/torque via capturing the spatial strain field of an elastic piezo-thermic material using web-like scattered thin-film thermistors on the top and bottom” in the revised manuscript.

The corresponding revision can be seen on Page 2-3 in the revised Manuscript.

Comment #3

The author designed a 5° tilt angle on the bottom of the sensor to “construct distinguishable spatial strain fields within the porous PDMS, aimed at identifying six-dimensional forces and torques”. However, it appears that even without the tilt, the strain fields in various regions could still be well differentiated. Please give a more detailed elaborate on the function of the tilt. Additionally, when M_z is applied (Fig. 2g), why does the principal strain exhibit a symmetrical distribution rather than forming a circular pattern aligned with the load direction? Please discuss

Response #3

Thanks for your comment.

We have added the simulation results of the spatial strain fields for the sensor without the bevel in the Supplementary Information (Figure S4). As you mentioned, if there is no bevel, M_z causes the strain cloud diagrams of the top and bottom layers to be circularly distributed, similar to the strain distribution caused by F_z . In this case, the sensor cannot differentiate F_z and M_z . Therefore, to enable the sensor to have the ability to distinguish six-dimensional forces and torques, we tailored this slight bevel. We further supplemented the simulation results at the bevel angles of 2.5° , 5° , and 7.5° in Figure S4g-i. As the bevel angle increases, the strain caused by M_z increases, which is beneficial to the measurement of torque. However, a large bevel angle is not conducive to the practical applications of the sensor. Considering the sensor performance and

practical application requirements, we chose a tilt angle of 5°.

The corresponding revision can be seen on Page 4 in the revised Manuscript and Page 7 in the revised Supplementary Information.

Comment #4

The author should further refine the calibration results of the sensor. In the supplementary tests shown in Fig. 3, why was only F_z tested? What about the responses of other forces or torques? Additionally, the response time test in Fig. 3h is not standardized when the change of F_z is less than 1% FS, which can not accurately reflect the dynamic characteristics of the sensor in actual use. Finally, in Fig. 3k, the coordinates lack numerical values, and most of the comparison subjects are not six-dimensional force sensors.

Response #4

Thanks for your comment and suggestion.

We have supplemented the detection limits of six-axis force/torque in the Supplementary Information (Figure S9). The detection limits of F_x , F_y , F_z , M_x , M_y , and M_z are better than 0.052 N, 0.05 N, -0.004 N, 1.20 mN·m, 1.22 mN·m and 0.31 mN·m, respectively. The response time of each axis force/torque is almost the same because they depend on the response time of the sensor unit, so here we only show the response time of F_z .

We have modified Figure 3h to show the response time of the sensor under the stimulus of F_z at -1 N (>1% FS). The response time and recovery time are both 90 ms.

We have added the numerical scale in Figure 3k, in which we compared our work with other flexible multi-axis force sensors. Furthermore, we have supplemented more six-axis force sensors in Table S1 for comparison.

The corresponding revision can be seen on Page 6-7 in the revised Manuscript and Page 12 and 18 in the revised Supplementary Information.

Comment #5

During the controlling experiments of the tank and the robotic arm, the operator merely uses their fingers to manipulate the sensor. Typically, multiple forces or torques would be applied simultaneously. However, the signals detected often respond to only specific types of forces or torques. Is this due to the operator's exceptionally precise hand control? When training the neural network, did the author use manual input or a calibration platform to generate the input signals? If the latter was used, then based on the platform setup shown in Fig. S5, it appears that the platform may not accurately measure the magnitude of torques.

Response #5

Thanks for your comment and suggestion.

In the process of manipulating the tank using sensors, in processes (i)-(iv) and (vii), the

operator's finger is located at the center of the sensor without torsion. At this time, the sensor mainly detects F_x , F_y , and F_z . In processes (v)-(vi), the operator's finger applies torque by adjusting the position of the applying force, and the sensor detects M_y and F_z simultaneously. The process of manipulating the robotic arm using the sensor is similar, so the sensor detects multiple forces and torques simultaneously and independently during the operation process (processes iii and vi in Figure 5b).

When training the neural network, we use a calibration platform (Fig. S6) to conduct force/torque experiments for data acquisition. We have added relevant schematic diagrams in Figure S6c to show the calculation of three-axis torque. When the location of the force is not at the center of the sensor, a torque and a force are applied simultaneously, and the torque can be accurately calculated as shown in Figure S6c.

The corresponding revision can be seen on Page 9 in the revised Supplementary Information.

Comment #6

At the end of the article (Fig. S9), the author suggests that a laminated design could be employed to detect different objects. However, the newly added components are entirely a reuse of the previous work, which lacks innovation and further complicates the sensor structure.

Response #6

Thanks for your comment.

With these newly added components, the tactile sensor can implement multimodal perceptions of object thermal conductivity, temperature, slippage, texture, and six-axis force/torque simultaneously. Such a multimodal tactile sensor is promising to achieve better human-robot interaction. And as far as we know, there is no sensor achieving this multimodal perception simultaneously. We agree that the design of multimodal perceptions should be oriented to the application requirements of tasks. For example, in a robotic dexterous manipulation, the robot needs not only six force/torque sensing, but also slipping detection and object recognition, so as to stably grasp objects and make fine sorting.

The corresponding revision can be seen on Page 11 in the revised Manuscript.

Thanks again for your valuable comments and suggestions.

Reviewer #2

The manuscript proposes a flexible six-axis force/torque sensor via capturing the spatial strain field of an elastic material utilizing scattered thin-film thermoreceptors. The sensor is ultralight (0.3 g), tiny, simple-structured, and the sensing performance is great in perceiving six-dimensional force/torque. The sensor particularly fits with the fingertip use, allowing dexterous fingertip manipulations. The manuscript also showcases some demonstrations of human-machine interaction tasks using the sensor. The work is interesting, the sensing design is novel, and the application is promising. There are minor issues that need to be addressed:

Comment #1

How does the environment temperature influence the sensor? How to realize temperature compensation?

Response #1

Thanks for your comments.

In Figure 3j and S10, we show responses of the sensor in the range of 25~45 °C, and the results indicate that the sensor is almost unaffected by the ambient temperature. Furthermore, we have added the detailed principle of temperature compensation in the Supplementary Notes. Through a Wheatstone bridge of the constant temperature difference circuit in Figure 1c and S3, a temperature difference ΔT between the hot-film and the environment is constant and not affected by the ambient temperature. Therefore, the sensor can accurately measure the six-axis force/torque at different temperatures.

The corresponding revisions can be seen on Page 2-3 in the revised Supplementary Information.

Comment #2

How about the humidity influence to the sensor?

Response #2

Thanks for your comment.

For humidity, since the thermal conductivity of water vapor is close to that of dry air, the thermal conductivity of moist air is about 0.0254~0.0255 W · m⁻¹ · K⁻¹ (relative humidity from 0 to 100%) (Desalination and Water Treatment, 2013, 51(4-6): 1290-1295). Therefore, humidity has almost no effect on the thermal conductivity of the air and therefore has no effect on the sensor's measurements. Furthermore, we have added an experiment result of the humidity effect on the sensor (relative humidity from 15% to 90%) in Figure S11, which indicates the sensor is immune from the humidity.

The corresponding revisions can be seen on Page 14 in the revised Supplementary Information.

Comment #3

According to the coordinate system defined in Figure 3b, I think F_z in Figure 3h~3j should be <0 . How many repeat measurements were taken in the experiment of Figure 3j? How were the error bars calculated?

Response #3

Thanks for your comment and suggestion.

As you said, F_z in Figure 3h~3j should be <0 . We have modified the relevant content in Figure 3h~3j.

Error bars shown in Figure 3j are standard deviations of five repeated measurements of one specific tactile sensor. We have added relevant explanations in the figure legend. The corresponding contents and revisions can be seen on Page 6-7 in the revised Manuscript.

Comment #4

How to train the data fusion model for figuring out six forces and torques needs to be described in more detail.

Response #4

Thanks for your comment.

The dataset contains 418 examples, where 40% of the dataset is the training set, 10% of the data is the validation set, and 50% of the dataset is the test set. The neural network is trained using the Bayesian regularized back-propagation method, with the learning rate set to 0.01 and the batch size set to 167.

We have added the above information on Page 12 in the revised manuscript.

Comment #5

What is the power consumption of the sensor?

Response #5

Thanks for your comment.

The power consumption of the sensor is about 80 mW.

We have added this information on Page 3 in the revised Manuscript.

Comment #6

The detailed operation of robot in the application of helping disabled people should be described more clearly.

Response #6

Thanks for your suggestion.

We have revised the description of the relevant content in the manuscript. For severely disabled people, steering electric wheelchairs is generally difficult. Using the six-axis force/torque sensor, disabled people can easily and flexibly steer the wheelchair with just one finger, which can provide great convenience.

The corresponding revision can be seen on Page 8 in the revised Manuscript.

Comment #7

In the first paragraph of Introduction, the piezoelectric principle also needs to be included for force sensors.

Response #7

Thanks for your suggestion.

We have added piezoelectric force sensors to the Introduction, including Ref. 8, 25, and 26.

The corresponding contents and revisions can be seen on Page 2 in the revised Manuscript.

Thanks again for your valuable comments and suggestions.

Reviewer #3

This work presents a flexible tactile sensor achieving 6-axis force/torque measurement through a thermoelectric-coupled design, addressing the challenge of high-dimensional signal decoupling measurement, demonstrating potential in delicate robot manipulation and human-mechanism interfaces. The study insufficiently validates flexibility, but is limited to flat-surface testing, lacks spatial resolution for multi-point force detection contradicting spiderweb-inspired claims, and omits many details of parametric optimization. Durability and environmental robustness remain unproven. Mandatory revisions are required to meet the publishing requirement including (1) sensor characterization under bending conditions (2) validation of localized/multi-force discrimination akin to spiderweb mechanics, (3) systematic parameter studies, and (4) expanded durability/environmental testing. Therefore, a major revision is recommended - methodological gaps must be resolved to solidify scientific contribution and translational value.

Response:

Thank you very much for your comments and suggestions. We have carefully revised the manuscript according to your suggestions. Specifically, we have validated the flexibility and characterized the sensor under bending conditions in Fig. S12, and added the corresponding parametric optimization for bevel (Fig. S4) and material ratio (Fig. S14). We have also validated durability (100,000 cycles with -15 N in Fig. 3i) and environmental robustness (almost no effect by environment temperature Fig. S10 and humidity Fig. S11). In addition, we have modified the spider-web-inspired expression to avoid the confusion of biometric comparison. More detailed responses are listed in the following.

Comment #1

Q1. What are the noteworthy results?

1. This paper presents a flexible 6-axis force sensor with compact dimensions that simultaneously measures triaxial forces (F_x , F_y , F_z) and torques (τ_x , τ_y , τ_z). This resolves the contradiction between high flexibility and high-dimensional signal decoupling in measurement.
2. The transformed mechanical stimuli into anisotropic thermal conduction signals (via Ag-doped porous PDMS), enabling unique responses to complex loads without bulky embedded electronics.
3. Demonstrated the implementation of the sensor into the robotic gripper's force sensing and remote robot control. The control of the gaming tanker and robotic arm are similar applications. The only difference is the first is a virtual robot while the latter is a physical robot.

Response #1

Thanks for your positive comments on our work.

Comment #2

Q2. Will the work be of significance to the field and related fields?

1. This work presents a flexible 6-axis force sensor. The proposed sensor also has the potential to provide a new way for human-machine interaction.

Response #2

Thanks for your comments.

Comment #3

Q3. How does it compare to the established literature? If the work is not original, please provide relevant references.

1. There are two concerns: the “flexible” claim is insufficiently supported by the experiments and analysis, and the spiderweb-inspired spatial perception is not mechanistically connected to the thermal-resistive design.

Response #3

Thanks for your comments. According to your suggestion in Q4, we have added the “flexible” related experiment result and analysis in Supplementary Notes and Figure S12 in the revised Supplementary Information. The results indicate that the thermal responses on the flat surface and the curved surface are basically the same, and the impact of bending can be ignored. As for spider-web-inspired spatial perception, spiders determine the location of prey on their webs by sensing spatial vibrations in their webs. Our sensor detects spatial strains to determine six-axis force/torque. The similarity is both spiders and our sensor perceive the spatial field of vibrations/strains to determine the target parameter. To avoid confusion, we have eliminated the biomimicry expression (Fig. 1c) and modified the description as “The sensor perceives the force/torque via capturing the spatial strain field of an elastic piezo-thermic material using web-like scattered thin-film thermistors on the top and bottom” in the revised manuscript.

The corresponding revision can be seen on Page 2-3 and 6 in the revised Manuscript and Page 2-3 and 15 in the revised Supplementary Information.

Comment #4-1

Q4. Does the work support the conclusions and claims, or is additional evidence needed?

1. Insufficient experiments: Force/torque tests were conducted on flat surfaces (Fig. 3a-b). However, no data on performance under bending conformal states are presented (e.g., wrapping around 10 mm radius curvatures), which is essential to validate “flexible” claims. The sensor's thermoelectric response may degrade significantly under non-planar deformation due to altered pore geometries and Ag particle connectivity. Therefore, more experiments and analysis under bending conditions are

required.

Response #4-1

Thanks for your comments and suggestions.

To minimize the impact of bending strain on the sensor, we design the hot-film and the cold-film with similar geometric shapes (Figure 1c). Combined with a constant temperature difference circuit, the impact of bending strain on the sensor signals can be eliminated. We have added relevant theoretical details for bending compensation in the Supplementary Notes.

Furthermore, we have added the sensor's response on a flat surface and a curved surface with a radius of 10 mm in the Supplementary Information (Figure S12). The results indicate that the thermal responses on the flat surface and the curved surface are basically the same, the impact of non-planar deformation on altered pore geometries and Ag particle connectivity can be negligible.

The corresponding revisions can be seen on Page 6 in the revised Manuscript and Page 2-3 and 15 in the revised Supplementary Information.

Comment #4-2

2. Overstated bio-inspiration: While spiderwebs excel in localized force detection and multi-point vibration discrimination, the current sensor only provides global 6-axis force estimation. There is no evidence of spatial resolution or multi-region independent sensing. It is suggested to apply two separate forces (e.g., F_x at the left edge and F_y at the right edge) to check if the sensor can resolve their locations and magnitudes simultaneously.

Response #4-2

Thanks for your comment and suggestion.

The sensor in this work aims to perceive six-axis force/torque on a finger, it is not for resolving multi-point force locations. As we have explained why we claimed bio-inspiration, both spiders and our sensor perceive the spatial field of vibrations/strains to determine the target parameter. Now we have eliminated the biomimicry expression (Fig. 1c) in the revised manuscript to avoid the confusion of biometric comparison.

The corresponding revisions can be seen on Page 2-3 in the revised Manuscript.

Comment #4-3

3. Not detailed parametric optimization: No theoretical/experimental proof for selecting a 5° tilt angle over other angles (e.g., 10°), and the justification for silver-doped porous polydimethylsiloxane, such as 2.5 vol% doping, is Vague.

Response #4-3

Thanks for your comment and suggestion.

We have supplemented the parametric optimization of 5° bevel and the strain results at bevel angles of 2.5°, 5°, and 7.5° in Figure S4(g)-(i). As the bevel angle increases, the strain caused by M_z increases, which is beneficial to the measurement of torque. However, a large bevel angle is not conducive to the practical applications of the sensor. Considering the sensor performance and practical application requirements, we choose a tilt angle of 5°.

We have supplemented sensor outputs when the volume ratios of silver nanoparticles are 1.5%, 2.5%, and 3.5% in the Supplementary Information (Figure S14). As the volume ratio of silver nanoparticles increases from 1.5% to 3.5%, the thermal conductivity of porous material gradually increases, which improves the sensitivity of the sensor. At the same time, since Young's modulus of porous materials decreases with the increase of the volume ratio of silver, the measuring range of the sensor gradually decreases. Therefore, to balance the sensitivity and range of the sensor, we choose the volume ratio of silver nanoparticles at 2.5%.

The corresponding revisions can be seen on Page 4 and 11 in the revised Manuscript and Page 7 and 17 in the revised Supplementary Information.

Comment #4-4

4. Incomplete model validation: CNN Generalization: Training/validation data (60/40 split) lack an independent test set. Crosstalk errors under combined loads (e.g., $F_x + \tau_z$) are unexplored.

Response #4-4

Thanks for your comment and suggestion.

We adjust the partition of the dataset, where 40% of the dataset is the training set, 10% of the data is the validation set, and 50% of the data is the test set.

We have supplemented the tested crosstalk errors among the forces/torques in the Supplementary Information (Figure S8). The results indicate that the crosstalk errors are low, which ensures simultaneous and independent perceptions of six-axis force/torque.

The corresponding revisions can be seen on Page 6 and 12 in the revised Manuscript and Page 11 in the revised Supplementary Information.

Comment #4-5

5. Insufficient durability test: 1000 cycles of 2 N loading (Fig. S3) are insufficient for medical/industrial use (>100,000 cycles needed per ISO 13485). Why not 5N and 15N, as the force-sensing range is 5N and 15N?

Response #4-5

Thanks for your comment and suggestion.

We have added the durability test results under 100,000 cycles of -15 N (Figure 3i). After 100,000 cycles of -15 N loading and unloading, the output of the sensor remains

stable, indicating that the sensor has good durability.
The corresponding revisions can be seen on Page 6-7 in the revised Manuscript.

Comment #5

Q5. Are there any flaws in the data analysis, interpretation, and conclusions? Do these prohibit publication or require revision?

Add more comparisons with other 6DOF sensors e.g., elastomeric strain gauges or FBG references.

Response #5

Thanks for your comment and suggestion.

We have added more experiment data and analysis to validate the merits of our sensor. We have also added a comparison with more six-axis force sensors based on different working principles including capacitive, visual, FBG, and strain gauges in Table S1 in the Supplementary Information.

The corresponding revisions can be seen on Page 18 in the revised Supplementary Information.

Comment #6

Q6. Is the methodology sound? Does the work meet the expected standards in your field?

Extension works are required as stated in Q4.

Response #6

Thanks for your comment.

We have responded Q4 in the above Responses.

Comment #7-1

Q7. Is there enough detail provided in the methods for the work to be reproduced?

1. Insufficient Protocol Details: Porous PDMS Fabrication: "Saccharine templating" (Methods) lacks critical parameters (e.g., stirring rate, humidity control). It is suggested to provide a step-by-step video protocol for PDMS-Ag layer synthesis.

Response #7-1

Thanks for your comment and suggestion.

In the porous PDMS fabrication, we use a blender (HVM200D, Hasai Technology Co. Ltd., Shenzhen, China) to blend the mixture thoroughly in three steps: (a) Revolution 900 rpm and rotation 600 rpm, 30 s. (b) Revolution 1500 rpm and rotation 600 rpm, 540 s. (c) Revolution 900 rpm and rotation 600 rpm, 30 s. During this fabrication, if heating is not required, the fabrication is at room temperature (23~25 °C) and humidity (10%~30%). We have revised the Methods in the revised Manuscript and Figure S5 in

the revised Supplementary Information, which ensures the reproduction of this fabrication process.

The corresponding revisions can be seen on Page 11-12 in the revised Manuscript and Page 8 in the revised Supplementary Information.

Comment #7-2

2. Neural Network Training: Missing hyperparameters (batch size, optimizer settings), raising barriers to replication. Open source the CNN codebase and raw calibration data will be appreciated.

Response #7-2

Thanks for your comment and suggestion.

The neural network is trained using the Bayesian regularized back-propagation method, with the learning rate set to 0.01 and the batch size set to 167. In addition, we have open-sourced the neural network code and raw calibration data on GitHub at <https://github.com/mq-0109/An-ultralight-tiny-flexible-six-axis-force-torque-sensor-enables-dexterous-fingertip-manipulations>.

The corresponding revisions can be seen on Page 12 in the revised Manuscript.

Thanks again for your valuable comments and suggestions.

Reviewer #4

Comment

Response

Thanks for your comments and suggestions.

All of the above revisions have been marked in **red font** in the text of the revised Manuscript and revised Supplementary Information.

If you have any other questions about this paper, please feel free to contact us.

Thanks for the comments again.

Sincerely

Rong Zhu, Ph.D.
Department of Precision Instrument
Tsinghua University, Beijing, China 100084
Tel: +86-010-62788935
E-mail: zr_gloria@mail.tsinghua.edu.cn

Dear Reviewers,

Thank you very much for your comments and efforts contributed to our paper. We have carefully investigated all comments. They are very helpful to improve the manuscript. We believe this has led to the substantial revision of the paper. The responses to these comments are listed in the following.

COMMENTS TO AUTHOR:

Reviewer #1

Comment #1

The author has addressed all my comments, thus this paper can be accepted.

Response #1

Thanks for your comments.

We greatly appreciate all the comments and efforts you provided during the review process.

Reviewer #2

Comment #1

The authors have revised the manuscript carefully according to the reviewer's comments. The quality of this work is greatly improved. The reviewer is satisfied with the revision and has no more other suggestion for this work. I think that this manuscript should be accepted by the journal of Nature Communications.

Response #1

Thanks for your comments.

We greatly appreciate all the comments and efforts you provided during the review process.

Reviewer #4

Comment #1

Please carefully check Fig. 3 (j). The results in this version and the previous version are opposite. Why? Please detail the reason.

Please carefully check Fig. 3 (j). The results in this version (left) and the previous version (right) are opposite. Why? Please detail the reason.

Response #1

Thanks for your comments.

In the first round of review, the Reviewer #2 pointed out “According to the coordinate system defined in Figure 3b, I think F_z in Figure 3h~3j should be <0 ”. After careful checking, we found our mistake of using $F_z > 0$ in Figure 3h~3j to represent the normal force in the z direction, which was inconsistent with the coordinate system defined in Figure 3b. Therefore, we corrected Figure 3h~3j in the previously revised manuscript. We defined $F_z < 0$ to represent the normal force in the z direction, which was consistent with the coordinate system in Figure 3b. Because of this reason, the normal force F_z in Figure. 3j should be negative, which was the opposite of that in the original version of the manuscript. But it is correct and consistent with the coordinate system we defined.

Thanks again for your valuable comments and suggestions.

If you have any other questions about this paper, please feel free to contact us.

Thanks for the comments again.

Sincerely

Rong Zhu, Ph.D.

Department of Precision Instrument

Tsinghua University, Beijing, China 100084

Tel: +86-010-62788935

E-mail: zr_gloria@mail.tsinghua.edu.cn

Please carefully check Fig. 3 (j). The results in this version (left) and the previous version (right) are opposite. Why? Please detail the reason.